# Electrochemical and Fluorescent Properties of Crown Ether Functionalized Graphene Quantum Dots for Potassium and Sodium Ions Detection

**DOI:** 10.3390/nano11112897

**Published:** 2021-10-29

**Authors:** Daniela Iannazzo, Claudia Espro, Angelo Ferlazzo, Consuelo Celesti, Caterina Branca, Giovanni Neri

**Affiliations:** 1Department of Engineering, University of Messina, Contrada Di Dio, I-98166 Messina, Italy; angelo.ferlazzo@unime.it (A.F.); ccelesti@unime.it (C.C.); gneri@unime.it (G.N.); 2Department of Mathematical and Computer Sciences, Physical Sciences and Earth Sciences, University of Messina, I-98166 Messina, Italy; cbranca@unime.it

**Keywords:** graphene quantum dots, K^+^ detection, Na^+^ detection, fluorescent sensors, electrochemical sensors

## Abstract

The concentration of sodium and potassium ions in biological fluids, such as blood, urine and sweat, is indicative of several basic body function conditions. Therefore, the development of simple methods able to detect these alkaline ions is of outmost importance. In this study, we explored the electrochemical and optical properties of graphene quantum dots (GQDs) combined with the selective chelating ability of the crown ethers 15-crown-5 and 18-crown-6, with the final aim to propose novel composites for the effective detection of these ions. The results obtained comparing the performances of the single GQDs and crown ethers with those of the GQDs-15-crown-5 and GQDs-18-crown-6 composites, have demonstrated the superior properties of these latter. Electrochemical investigation showed that the GQDs based composites can be exploited for the potentiometric detection of Na^+^ and K^+^ ions, but selectivity still remains a concern. The nanocomposites showed the characteristic fluorescence emissions of GQDs and crown ethers. The GQDs-18-crown-6 composite exhibited ratiometric fluorescence emission behavior with the variation of K^+^ concentration, demonstrating its promising properties for the development of a selective fluorescent method for potassium determination.

## 1. Introduction

Sensors for the selective quantification of sodium and potassium ions are of crucial importance in clinical diagnostics [1,2,3]. These ions are essential electrolytes for human homeostasis and changes in their concentrations can be related to serious complications for health, such as compromised heart function [4], cystic fibrosis [5], kidney failure [6], diabetic ketoacidosis [7] and adrenal disorders [8]. Sodium is the most abundant cation present in extracellular fluids, with a concentration rate between 135 and 145 mEq/L, while potassium represents the most abundant cation inside cells, with concentrations in the range of 140–150 mEq/L [9]. These two electrolytes have been described as the body’s dynamic duo, since they work in concert for energy production and to maintain the body’s fluid balance [10]. Molecular sodium-potassium pumps pull potassium into cells and push sodium out, thus creating a chemical battery that is able to drive nerve signal transmission and to activate muscle contraction. The evaluation of the concentrations of these electrolytes, not only in serum, but also in sweat and in dermis, can provide important information on the human body’s hydration status [11,12]. Their quantification can provide important information for therapeutic interventions in elderly people, especially in clinical or nursing home settings, and for sports performance optimization. Thus, their precise, sensitive and, if possible, contemporaneous monitoring, could allow the early diagnosis of various physiological and pathological events and to track the efficacy of medical treatments for correct therapeutic interventions.

Even if the existing techniques allow the sensitive quantification of these electrolytes in blood and urine samples, their use is limited by their cost, speed and the large amount of sample required [9,13]. Moreover, the different existing analytical methods for their detection do not allow for continuous monitoring in remote and in resource-limited environments. Thus, easy, sensitive, rapid, and low-cost methods for the real-time tracking of Na^+^ and K^+^ concentration in biological fluids are required to ensure the diagnosis and medical treatment of this fast-moving disease. Several studies on the sensitive and rapid detection of these electrolytes in the biological fluids have been proposed, including the use of wearable electrochemical sensors based on conductive polymers [12,14], extended gate field effect transistor for multi-ions sensing [15], ion-selective optical sensors [16,17] and paper-based biosensors [18].

However, the achievement of a selective response in Na^+^ or K^+^ detection is still a challenge. One of the reasons for the difficulty of distinguishing the signals of the two ions lies in their similar chemical properties. The possibility to selectively detect these two ions can be achieved by taking advantage of their different Pauling ion radii (1.02 Å for sodium and 1.38 Å for potassium) [19]. Among the different supra-molecular host molecules able to form stable host-guest complexes with ions in function of their size, crown ethers represent the standard due to their stronger affinities for alkali metal cations in solution [20]. Their cation selectivity is dependent on the size and charge density of the ion and the crown ether cavity and is mainly achieved by ion-dipole interactions [21]. Among them, 15-crown-5 and 18-crown-6 are good hosts for sodium and potassium ions, respectively, and have been investigated for the development of selective fluorescent, potentiometric and electrochemical sensors for the detection of these biologically important electrolytes in solution [22,23,24,25,26].

Graphene based materials, due to their high electrical conductivity and large planar area, exhibit favorable properties for biosensing and bioimaging applications [27,28,29,30]. Moreover, recent theoretical studies suggest that the existing crown ether configurations in graphene could afford a much larger binding energy [30]. A graphene-based fluorescence resonance energy transfer (FRET) sensor linked to an ion-selective crown ether revealed a detection limit of K^+^ ion of about 1 × 10^−5^ M [23]. Reduced graphene oxide (rGO) covalently conjugated with 18-crown-6 ether displayed the ability to detect K^+^ ions down to the micromolar range [25], while an electrochemical sensor based on 1-aza-18-crown-6 functionalized GO reported a detection limit for K^+^ ion of 10^−15^ M [26]. The next generation of the graphene family, the zero dimensional graphene quantum dots (GQDs), received significant interest within academia and industry over the last few years due to their remarkable physicochemical properties [31,32]. Unlike two-dimensional graphene, GQDs have a band gap because of the quantum size effect and exhibit stable and size-dependent photoluminescence, which allows for their application in photovoltaics, electro/photo/chemical catalysis, the fabrication of flexible devices and in biosensing [33,34,35,36].

In this study, we report, for the first time, the combination of the electrical and optical properties of GQDs and the selective chelating ability of the crown ethers 15-crown-5 and 18-crown-6 with the aim of developing sensitive electrochemical and fluorescent sensors for sodium and potassium ions. The covalent conjugation of GQDs with the selected crown ethers allows for the development of nanocomposites able to discriminate between the two ions because of their different binding mechanisms. The results of this study show that the electrochemical properties and the fluorescence behavior of these composite nanomaterials make possible and promising the development of simple, rapid, and effective tests for the detection of Na^+^ and K^+^ ions, with the fluorescent method also able to achieve a selective detection towards potassium ions.

## 2. Materials and Methods

### 2.1. Materials

The chemical reagents and solvents employed in this study were purchased from commercial suppliers and used without any further purification. GQDs were obtained from pristine multi-walled carbon nanotubes (MWCNTs), following a previously reported procedure [37]. Briefly, MWCNTs were treated with a mixture of HNO_3_/H_2_SO_4_ (1:3 ratio) in an ultrasonic bath at 60 °C for four days. The suspension was diluted with deionized water and filtered on a 0.1 µm Millipore membrane. The filtrate was treated with a solution of NaOH until it reached a neutral pH and was purified using dialysis bags (12 KDa). The number of acidic groups, as evaluated by titration analysis, was found to be of 2.37 mmol/g.

### 2.2. Synthesis of GQDs-15-Crown-5 and GQDs-18-Crown-6

The composite nanomaterials were synthesized as follows. A water solution of GQDs (30 mg/30 mL) was added to 10 mL of dimethylformamide (DMF) and, after removal of water under reduced pressure, the mixture was treated with N-(3-dimethylaminopropyl)-N-ethylcarbodiimide hydrochloride (EDC·HCl, 0.14 mmol), triethylamine (ETA, 0.14 mmol) and left to stir at room temperature for 15 min. Hydroxyl benzotriazole (HOBt, 0.14 mmol) and a catalytic amount of 4-dimethylaminopyridine (DMAP) were then added and the mixture was left to stir for an additional 1 h. The suspension was then treated with of 2-aminomethyl-15-crown-5 (0.12 mmol) for GQDs-15-crown-5 synthesis or 2-aminomethyl 18-crown-6 (0.1 mmol) for GQDs-18-crown-6 synthesis and left to stir at room temperature for four days. The obtained suspensions were then diluted with deionized water and purified using dialysis bags with MW of 12KDa until no organic material was present in the washing solutions. The degree of functionalization was evaluated on a known amount of sample (dried under vacuum at 60 °C) by thermogravimetric analysis, under argon atmosphere.

### 2.3. Microstructural, Optical and Electrochemical Characterizations

Thermogravimetric studies were carried out at 10 °C/min, from 100 to 1000 °C, in argon atmosphere using TGA Q500 (TA instruments). The size determination was performed by dynamic light-scattering (DLS) analysis using the Zetasizer 3000 instrument (Malvern), equipped with a 632 nm HeNe laser and operating at a 173-degree detector angle. Infrared spectra were registered using a Perkin Elmer Spectrum 100 spectrometer, equipped with a universal ATR sampling accessory; spectra were recorded without any preliminary treatment, at room temperature, from 4000 to 600 cm^−1^, with a resolution of 4.0 cm^−1^. Micro-Raman spectra were measured in VV backscattering geometry using a LabRam HR 800 spectrometer. The spectra were recorded with an excitation wavelength of a 532 nm semiconductor diode laser, a 50× objective and a 77 K cooled charged couple device detector.

Photoluminescence (PL) measurements were performed at room temperature using a spectrofluorometer NanoLog modular (Horiba), under excitation with a xenon lamp; the GQD based nanomaterials were analyzed at the concentration of 100 ng/mL. To test the electrochemical properties, screen-printed commercial electrodes (SPCE) were functionalized directly by drop casting 18 microliters of a dispersion (1 mg/mL) of the GQDs based nanomaterials on the working electrode of SPE, and then left to dry overnight at room temperature. Electrochemical experiments were performed using a DropSensμStat 400 potentiostat. The electrochemical behavior of the developed sensors was investigated in a deionized water solution at pH 7, in the presence and absence of KCl or NaCl, performing potentiometric measurements, i.e., measuring the potential difference between the two electrodes under the conditions of no current flow.

## 3. Results and Discussion

### 3.1. Synthesis and Characterization of GQDs-15-Crown-5 and GQDs-18-Crown-6

The synthesis and characterization of the initial GQDs, obtained by acidic oxidation and chemical exfoliation of MWCNTs, was previously reported [37]. These carbon nanodots, with a weighted size distribution of 4.8 nm, are characterized by the presence of many oxygen containing functional groups, due to the strong acidic treatment during their synthesis. The presence of carboxyl functionalities allows the covalent conjugation of GQDs with crown ethers functionalized with nucleophilic amino groups.

The coupling reaction between 2-aminomethyl-15-crown-5 (NH_2_-15-crown-5) or 2-aminomethyl-18-crown-6 (NH_2_-18-crown-6) to the surface of GQDs was performed by activating the carboxylic functionalities present on the graphene surface using EDC/HOBt and 0.1 equivalent of DMAP, thus leading to the formation of amide bonds between these groups and the crown ethers (Figure 1).

After the removal of unreacted reagents and reaction solvent by dialysis, the effectiveness of the coupling reactions leading to GQDs-15-crown-5 and GQDs-18-crown-6, as well as the amount of crown ethers moieties loaded onto GQDs, was investigated by Fourier transform infrared spectroscopy (FTIR) and by thermogravimetric analysis (TGA) (Figure 1). These characterization techniques are widely used to prove the effectiveness of organic functionalization for carbon based nanomaterials. In particular, FTIR spectroscopy allows for proving the covalent functionalization of organic molecules to the surface of graphene based materials. The FTIR spectrum of GQDs (Figure 1a) shows the presence of a peak at 1610 cm^−1^ due to the stretching of the C=O group of the carboxyl functionality and a large band at 3450 cm^−1^ related to the stretching of the O−H bond. These peaks are indicative of the presence of many oxygen-containing groups on the surface of GQDs. The spectra of the conjugated samples GQDs-15-crown-5 and GQDs-18-crown-6 show the additional representative peaks at 1120 cm^−1^ which can be assigned to the ether functional group (C–O–C) of crown ethers, while the additional peaks at 1640 cm^−1^ are attributable to the newly formed amide bonds between the crown ethers and the GQDs.

The TGA curves of the GQDs and of the corresponding crown ethers functionalized GQDs, performed under inert atmosphere, show an increase inweight loss for both conjugated samples, the amounts of which, as calculated at 500 °C, were found to be of 24 wt% and 35.1 wt% for GQDs-15-crown-5 and GQDs-18-crown-6, respectively (Figure 1b). Moreover, the different profiles of the TGA curves of the conjugated systems, confirm that the deep chemical modifications occurred on the nanomaterials after the functionalization processes.

Raman spectroscopy is a very useful technique forinvestigating the level of disorder in graphene based materials. Raman spectra of GQDs and of the crown ethers conjugated samples confirmed that the structural modification occurred on the conjugated samples after the functionalization procedures (Figure 2a). In particular, we have investigated the variations in the intensities of the D-band (ca. 1360 cm^−1^) and G-band (ca. 1590 cm^−1^) of the starting GQDs, GQDs-15-crown-5 and GQDs-18-crown-6 after exciting the samples at 532 nm. The D-band is related to local defects that originate from structural imperfections of the aromatic networks, while the G-band is considered as the Raman fingerprint of the graphitic crystalline arrangements [38]. The I_D_/I_G_ ratio is usually used to evaluate the presence of disorder in sp^2^ hybridized carbon systems and to prove the effectiveness of the functionalization reactions. This value, which is equal to 1.18 for the starting GQDs, decreased for both functionalized samples to 1.068 and 1.065 for GQDs-15-crown-5 and GQDs-18-crown-6, respectively. These values demonstrated that the functionalization procedures enhanced the crystalline structure of the sp^2^ hybridized carbon network.

To verify if the particle size of the GQDs have been maintained after the functionalization processes, we have investigated the conjugated samples by DLS (Figure 2b). This technique allows for evaluating the size distribution profile of small particles by measuring the random changes in the intensity of light scattered from the particles’ suspension. The volume-weighted DLS measurements, performed on dispersions of the conjugated samples in deionized water, showed single size populations centered at 4.93 and 4.88 nm, for GQDs-15-crown-5 and GQDs-18-crown-6, respectively. The observed values are superimposable with that of the starting GQDs, thus confirming that no particles aggregation occurred on the nanomaterials after the functionalization procedure.

### 3.2. Electrochemical Studies

The electrochemical characteristics of the nanocomposite samples have been investigated to evaluate their possible use as ionophores in membrane-free screen-printed potentiometric sensors. Crown ethers have been previously reported as ionophores in ion-selective electrodes (ISEs) [39]. Potentiometric tests have been performed in the Na^+^ and K^+^ concentrations range of 1 to 1000 mM in deionized water at pH 7. Figure 3 reports the behavior of GQDs-18-crown-6/SPCE towards K^+^ ions, by immersing the electrode in potassium ion solutions of different concentrations.

The electrode response time was estimated by recording the time needed to attain a steady state potential after a sudden tenfold increase in the metal ions concentration [40]. Composite based electrodes showed a fast response (less than 4s, see inset in Figure 3). The good sensing properties described may be attributed to the synergistic effect between the high electron transfer properties of GQDs and the chelating ability of crown ethers within the electrode matrix. This behavior was observed also using the GQDs-15-crown-5/SPCE sensor.

Figure 4 reports the calibration curve of GQDs-18-crown-6/SPCE for the detection of K^+^ ions. The comparison with the single GQDs and the 18-crown-6 based sensor confirms that the potentiometric response is largely enhanced by the presence of the GQDs-18-crown-6 composite as ionophore. Moreover, a linear response with the increase of the K^+^ concentration was recorded. The nanocomposite-based sensors showed a super-high Nernstian slope value (75.3 mV/decade). This anomalous high response has been already reported for chelating ionophores, such as crown ethers [41]. However, the data reported in the same plot in the presence of Na^+^ evidenced clearly that is not possible to discriminate between the two ions. The same findings, in terms of response and selectivity, were found for the GQDs-15-crown-5/SPCE sensor.

EIS measurements (Appendix A) provided further information helpful to understand the electrochemical sensing mechanism. The strong decrease in the charge transfer resistance for GQDs/SPCE reveals the more conductive nature and faster electron transfer rate as compared to the bare SPCE. These data demonstrated that the GQDs-18-crown-6/SPCE sensor benefits of the presence of GQDs in the composite formulation. Indeed, despite the low conductivity of 18-crown-6 moiety, the GQDs-18-crown-6/SPCE exhibits a good charge transfer rate, which is an important parameter forenhancing the performance of the electrochemical sensor. Combining EIS data with the potentiometric results, it appears that the complexation of Na^+^ and K^+^ ions occurring at the electrode-solution interface, facilitating the fast transfer into the electrode surface and the fast electron transfer, are at the origin of the good response of the GQDs-18-crown-6/SPCE to the changes of concentration of these ions in solution. To evaluate the stability of the GQDs-18-crown-6 based electrode, we performed repetitive measurements. We found that, for tests carried in successive days, the response of the electrode was nearly unchanged (Appendix A).

### 3.3. Photoluminescence Studies

The photoluminescence properties of the synthesized nanomaterials were investigated by comparing the PL properties of the starting nanomaterials with those of the corresponding crown ethers conjugated samples in deionized water at the excitation wavelength of 360 nm (Figure 5).

The GQDs-15-crown-5and GQDs-18-crown-6 samples show a strong emission peak with a maximum at 550 nm and a shoulder at around 450 nm. This additional emission peak at the lower wavelength can be reasonably attributed to the crown ether components (see PL of NH_2_-15-crown-5 and NH_2_-18-crown-6 samples in Appendix A), in agreement with results reported in the literature for similar crown ether conjugates [22,42]. The observed blue-shift of emissions of the conjugated samples with respect to the starting GQDs can be attributed to the covalent conjugation with the crown ethers [43].

The fluorescence behavior in the presence of different concentrations of Na^+^ and K^+^ ions was then investigated. Analogously to potentiometric studies, concentrations of the analytes in the range of 1 to 1000 mM in deionized water were used. The photoluminescence performances of single GQDs and the free crown ethers (Appendix A) were firstly evaluated. As shown in Appendix A, a slight increase in fluorescence was observed when increasing concentrations of the two ions were added to the GQDs solutions, regardless of the added salt. For the unbound crown ethers, the observed increase in emission (Appendix A) is particularly evident for the complexation of the ion of interest with the corresponding crown ether (K^+^ for NH_2_-18-crown-6 and Na^+^ for NH_2_-15-crown-5), as also reported in the literature for a similar system [44,45].

The photoluminescence performances of the conjugated samples GQDs-15-crown-5 and GQDs-18-crown-6 are shown in Figure 6. The GQDs-15-crown-5 sample shows a clear affinity to Na^+^ ions, affording enhancements of fluorescence intensity when increasing concentrations of this analyte were added to the system (Figure 6a); conversely, increasing additions of K^+^ ion to the same system did not exert consistent changes in the fluorescence emission (Figure 6b). For the GQDs-18-crown-6 sample, a different PL behavior was observed (Figure 6c,d). Enhancements of fluorescence were recorded at 450 nm after the addition of increasing concentrations of Na^+^ ion, while the addition of increasing concentrations of K^+^ to the same system afforded anincreasing quenching of fluorescence at 550 nm and increasing in PL emission at 450 nm.

Analogous photoluminescence behavior was observed for this sample, after addition of the same concentration (100 mM) of both ions (Appendix A). An increase in fluorescence was still recorded at 450 nm, together with a quenching of fluorescence at 550 nm, thus further confirming the PL mechanism of this sample in the presence of the K^+^ ion.

The increase in PL emissions at 450 nm and the contemporary quenching of fluorescence at 550 nm allows for proposing the composite GQDs-18-crown-6 for a ratiometric sensing method for potassium ions detection. Ratiometric sensing, based on the ratio of fluorescence intensities at two wavelengths, results are more sensitive and reliable than those from a single fluorescence probe because it canprovide a built-in self-calibration as an internal standard [46]. Carbon dots-based nanomaterials have, in the past, been used to develop similar ratiometric systems [47]. From the results reported in Figure 6c,d, we computed the I_450_/I_550_ intensity ratio in the presence of various concentrations of the alkaline ions and reported these values vs. the ions concentrations (Figure 7). The increase of fluorescence intensity at 450 nm and the contemporary decrease at 550 nm results in the consequential increase of the I_450_/I_550_ intensity ratio with an increase in the K^+^ concentration. Figure 7 also shows that the I_450_/I_550_ intensity ratio for Na^+^ increases less than for K^+^ in the same concentration range, thus suggesting that the GQDs-18-crown-6 composite displays a certain selectivity towards potassium ions.

This behavior can be explained by considering a different complexation mechanism between the GQDs-18-crown-6 composite and the K^+^ ion. As also reported in the literature for similar systems [16], this ion can interact with the oxygens of the crown ether and with the amide oxygens nearby the GQDs surface; the interaction with the oxygen containing groups of the GQDs surface is also expected. This conformational change stabilizes the complex with the K^+^ ion and increases the interaction of the crown ether with the GQDs surface. As also reported in other studies, the non-covalent interaction of small molecules with the GQDs surface can lead to the observed quenching phenomenon [48,49]. On the basis of the observed fluorescence behavior, we propose the fluorescence mechanism for the GQDs-18-crown-6 composite system in the presence of Na^+^ or K^+^ ions, as shown in Figure 2.

As is well evident, binding of both Na^+^ and K^+^ ions turn on the fluorescent peak related to the GQDs-18-crown-6. However, in the presence of K^+^ ions, quenching of the fluorescent peak related to GQDs moiety occurs too, thus enabling the ratiometric fluorescence sensing. This observation leads us to hypothesize that the interaction of Na^+^ with the oxygens of the crown ether and also with the amide oxygens nearby the GQDs surface, as supposed for K^+^, does not occur. This different behavior could include both steric and/or electronic factors. Regardless, it appears that GQDs serves as reference fluorophore, while the crown ether serves as a recognition unit with a turn-on response. We are extending this study to other ions to better understand the origin of the differences observed between the investigated alkaline ions.

## 4. Conclusions

Sensors for the selective quantification of sodium and potassium ions are of crucial importance in clinical diagnostics, since changes in the concentrations of these ions can be related to serious complications for human health. In this study, we have investigated the electrochemical and optical properties of GQDs and the selective chelating ability of the crown ethers 15-crown-5 and 18-crown-6, for the development of fluorescent and electrochemical sensors for the determination of these two ions.

The electrochemical performance of the synthesized nanocomposites, evaluated at the concentration range up to 1 mM, demonstrated high sensitivity; however, low selectivity between these ions was observed.

Both nanocomposites show the characteristic fluorescence emissions of GQDs and crown ethers under a single excitation wavelength. Interestingly, for the GQDs-18-crown-6 sample, a ratiometric fluorescence emission behavior with the variation of K^+^ concentration was observed.

The observed fluorescence behavior of this sample can allow for the development of simple, rapid, and effective tests for the selective detection of potassium over sodium ion. The extension of this study to other ions, also by electrochemiluminescence (ECL) tests, will allow for a better understanding of the origin of the differences observed between the investigated ions.

## Data Availability

The data presented in this study are available in the article.

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
