# Peer review of "Electrochemical and Fluorescent Properties of Crown Ether Functionalized Graphene Quantum Dots for Potassium and Sodium Ions Detection"

_nanomaterials, 2021, doi:10.3390/nano11112897_

Round 1
Reviewer 1 Report
In this paper, Iannazzo et.al. explored the electrochemical and optical properties of graphene quantum dots (GQDs) combined with the selective chelating ability of the crown ethers 15-crown-5 and 18-crown 6, with the final aim to propose novel composites for the effective detection of these ions. This work is interesting and I could recommend its publication in the Nanomaterials.
The major concerning is listed as followed:
- What is the reason why the PL intensity decreased when GQDs meets the ions?
- Did the crown ester affect the properties of GQDs after modification?
Author Response
Response to Reviewer 1:
In this paper, Iannazzo et.al. explored the electrochemical and optical properties of graphene quantum dots (GQDs) combined with the selective chelating ability of the crown ethers 15-crown-5 and 18-crown-6, with the final aim to propose novel composites for the effective detection of these ions. This work is interesting and I could recommend its publication in the Nanomaterials.
We thank the Reviewer for the time devoted in revising the manuscript and for the positive comments.
The major concerning is listed as followed:
- What is the reason why the PL intensity decreased when GQDs meets the ions?
Response: We thank the Reviewer for his valuable comment. The decrease in the PL intensity of GQDs was registered only with K+ ions. This effect was rationalized by the different complexation mechanism (with respect to that observed for Na+ ion) between the GQDs-18-crown-6 composite and the K+ ion. This mechanism, which involves the interaction of the ion with the oxygens of the crown ether and the amide oxygens nearby the GQDs surface was already reported in the literature (see ref. [16] of the manuscript). We believe that this different binding mechanism could increase the interaction of the crown ether molecules with the GQDs surface, thus leading to a quenching phenomenon. This quenching mechanism was already reported for other small molecules after their noncovalent interaction with GQDs. In order to better clarify this issue we have reported a brief discussion, before Scheme 2 and have added the new related references [48] and [49]. Moreover, we have further highlighted this aspect by adding arrows indicating the increase and decrease in photoluminescence in Figure 6.
- Did the crown ester affect the properties of GQDs after modification?
Response: As reported in literature, the structural modification of GQDs have shown to tune their physical, chemical, and biological properties (see Ref. [32] in the manuscript). In our study, we observed a blue-shift of emission for the crown ether functionalized nanomaterials with respect to the starting GQDs which was attributed to the covalent conjugation with the crown ethers. However, the electrochemical and fluorescence properties of the conjugated samples allowed us to discriminate between the two investigated ions. In particular, the observed fluorescence behavior of GQDs-18-crown-6 has shown to allow the selective detection of potassium over sodium ion.
Reviewer 2 Report
In this manuscript, the author reports, ‘Electrochemical and fluorescent properties of crown ether functionalized graphene quantum dots for potassium and sodium ions detection’. The current study is on a topic of relevance and general interest to readers in this area. The authors should address the following questions before getting a possible publication.
Recommendation: Major revisions needed as noted.
- The implication for attaching the crown ether to GQDs is not clear. It will be better if the author will discuss it.
- Did the author perform an electroluminescence test of normal PL? Make it clear.
- Could the author inform about the host-guest binding propensity when sodium and potassium both were utilized?
- The PL platform (GQDs) based works could be described in literature review by addressing the articles: Research on Chemical Intermediates7 (2019): 3823-3853 ; Luminescence33.6 (2018): 1136-1145; Talanta 117 (2013): 152-157.
- Did the author check how the sensitivity was changed with their ionic strength?
- The author should write the purpose for each test in one/two sentences (in brief) before explaining the results of the characterization techniques. Therefore, the logic and organization of this part will be enhanced.
- The formatting and grammatical errors in the article need to be checked carefully.
- What does the error bars stand for presented in the Figure 7? It should be mentioned in Figure captions.
Author Response
Response to Reviewer 2
In this manuscript, the author reports, ‘Electrochemical and fluorescent properties of crown ether functionalized graphene quantum dots for potassium and sodium ions detection’. The current study is on a topic of relevance and general interest to readers in this area. The authors should address the following questions before getting a possible publication.
Recommendation: Major revisions needed as noted.
We thank the Reviewer for the time devoted in revising the manuscript and for his valuable comments that helped us to improve the quality of the paper.
- The implication for attaching the crown ether to GQDs is not clear. It will be better if the author will discuss it.
Response: Aim of this study was to combine the electrical and optical properties of GQDs and the selective chelating ability of the crown ethers 15-crown-5 and 18-crown 6 towards sodium and potassium ions, respectively. These crown ethers can distinguish the signals of the two ions by taking advantage of their different Pauling ion radii (see Ref. [19] in the manuscript) and have been already investigated for the development of selective fluorescent, potentiometric and electrochemical sensors (see Refs. [22‒26] in the manuscript). The covalent conjugation of GQDs with the selected crown ethers allowed us to develop new nanocomposites able to discriminate between the two ions, due to their different binding mechanism. In order to better clarify this issue, we have added a sentence about the rationale of our approach, at the end of the Introduction section.
- Did the author perform an electroluminescence test of normal PL? Make it clear.
Response: We really thank the reviewer for his valuable comment. Recently, particular interest has been devoted to the electrochemiluminescence (ECL) performance of GQDs, due to their promising use in biosensing and bioimaging. We have not performed ECL tests in this study, but the future objective of this research work will be the evaluation of luminescent signals by ECL for the synthesized composites, also extending this study to other ions. We have added the possible extension of this study, involving ECL tests, in the Conclusions section.
- Could the author inform about the host-guest binding propensity when sodium and potassium both were utilized?
Response: We thank the reviewer for his precious suggestion. We have reported the host-guest binding ability of the synthesized composites in the presence of both ions by PL measurements. We still observed an increase in fluorescence at 450 nm together with a quenching of fluorescence at 550 nm, thus further confirming the PL mechanism of this sample in the presence of K+ ion. We have discussed these data in the results and discussion section and have reported the related PL spectra in the supplementary materials (Figure S4).
- The PL platform (GQDs) based works could be described in literature review by addressing the articles: Research on Chemical Intermediates 7 (2019): 3823-3853 ; Luminescence 33.6 (2018): 1136-1145; Talanta 117 (2013): 152-157.
Response: We thank the reviewer for the suggested literature data, related to the photoluminescence properties of GQDs and have added the above reported references (Refs [34-36] in the manuscript).
- Did the author check how the sensitivity was changed with their ionic strength?
Response: We agree with the Reviewer in considering that the ionic strength is an important factor that could affect the photoluminescence behavior of materials used for optical sensing. However, in our experiments, at the tested concentrations of the investigated salts, the effect of ionic strength was not observed. Even when the concentrations of salts were up to 1000 mM and then higher than the physiological ionic strength (~100 mM), the PL intensity still was almost unchanged. This observation is in perfect agreement with similar studies regarding the dependence of photoluminescence with ionic strength in biosensing (see Helin Liu et al., Scientific Reports, 2015, 5, 14879).
- The author should write the purpose for each test in one/two sentences (in brief) before explaining the results of the characterization techniques. Therefore, the logic and organization of this part will be enhanced.
Response: We thank the reviewer for the precious suggestion which allowed us to improve the quality of the manuscript. We have added a brief discussion regarding the rationale of the used technique for the reported characterizations and in particular, for FTIR, Raman and DLS measurements.
- The formatting and grammatical errors in the article need to be checked carefully.
Response: We have checked the manuscript and corrected the punctuation and grammar errors.
- What does the error bars stand for presented in the Figure 7? It should be mentioned in Figure captions.
Response: We thank the reviewer for having highlighted this mistake. The error bars in Figure 7 indicate the standard deviation of the experiments, done in triplicate. We have added this information in the legend of Figure 7.
Round 2
Reviewer 1 Report
The authors has addressed all the issues. I could recommend its publication.
Reviewer 2 Report
The authors have addressed all the questions raised before. The manuscript can be accepted in the present form.